# Deterministic transfer of optical-quality carbon nanotubes for atomically defined technology

Keigo Otsuka [1,5✉], Nan Fang [1], Daiki Yamashita [2], Takashi Taniguchi [3], Kenji Watanabe [4] & Yuichiro K. Kato [1,2✉]

When continued device scaling reaches the ultimate limit imposed by atoms, technology based on atomically precise structures is expected to emerge. Device fabrication will then require building blocks with identified atomic arrangements and assembly of the components without contamination. Here we report on a versatile dry transfer technique for deterministic placement of optical-quality carbon nanotubes. Single-crystalline anthracene is used as a medium which readily sublimes by mild heating, leaving behind clean nanotubes and thus enabling bright photoluminescence. We are able to position nanotubes of a desired chirality with a sub-micron accuracy under in-situ optical monitoring, thereby demonstrating deterministic coupling of a nanotube to a photonic crystal nanobeam cavity. A cross junction structure is also designed and constructed by repeating the nanotube transfer, where intertube exciton transfer is observed. Our results represent an important step towards development of devices consisting of atomically precise components and interfaces.

[1] Nanoscale Quantum Photonics Laboratory, RIKEN Cluster for Pioneering Research, Saitama, Japan. [2] Quantum Optoelectronics Research Team, RIKEN Center for Advanced Photonics, Saitama, Japan. [3] International Center for Materials Nanoarchitectonics, National Institute for Materials Science, Ibaraki, Japan. [4] Research Center for Functional Materials, National Institute for Materials Science, Ibaraki, Japan. [5] Present address: Department of Mechanical Engineering, The University of Tokyo, Tokyo, Japan. ✉email: otsuka@photon.t.u-tokyo.ac.jp; yuichiro.kato@riken.jp

"There's plenty of room at the bottom," Richard Feynman said in his 1959 lecture[1]. Since the first notion of nanotechnology, device miniaturization has been the driving force for technological evolution. The rapid progress of nanofabrication techniques has pushed forward Moore's Law, exponentially increasing the computation powers of semiconductor microprocessors that now form the foundations for large-scale simulations and artificial intelligence. Fueled by scientific curiosity and technological thirst, efforts for further scaling will undoubtedly continue until we reach the ultimate limit where devices are constructed of components and interfaces with atomic precision: Atomically defined technology.

Recent developments in heterostructures of two-dimensional (2D) materials[2–4] have highlighted novel exotic phenomena arising from the intricacies of precise atomic arrangements. Twisted bilayer graphene exhibits superconductivity at the magic angle[5], whereas twisted homobilayer of 2D semiconductor with a long-period moiré superlattice gives rise to incompressible Mott-like states of electrons[6]. In addition, high-temperature condensation of interlayer excitons has been observed in aligned $MoSe_2$–$WSe_2$ atomic double layers[7], where control over the interlayer spacing by atomically thin insulators also plays a key role.

To harness the full functionality and performance of atomically precise structures, systems with more complexity than stacked layers would be necessary. It is, however, not an easy task to assemble atomically defined building blocks for real-world applications. Organic molecules allow for tailoring their structures at the atomic level[8,9], but manipulation of single molecules requires low temperature and ultrahigh vacuum[10]. Although it is possible to integrate individual semiconductor nanostructures into electronic and photonic devices[11,12], it still remains a challenge to reproducibly prepare identical structures with atomic precision.

Carbon nanotubes (CNTs) offer a unique position in this context. Their atomic arrangements can be specified by chirality which is a combination of two integers defining the geometry of the roll-up vector, and they can be individually addressed to construct various nanoscale devices such as single-electron transistors [13] and light-emitting diodes[14]. Simultaneous control over chirality, position, and orientation of CNTs would enable the fabrication of devices utilizing atomically precise components, given that their intrinsic properties are preserved during assembly. Picking up air-suspended tubes and re-suspending them is an ideal method for keeping the surface pristine[15–18], but the necessity for three-dimensional structures imposes limitations on device construction. In addition, state-of-the-art growth control over tube diameter and length on substrate surfaces[19–21] cannot be utilized. A different approach for manipulating chirality-identified tubes with negligible contamination would be required to achieve the flexibility needed for integrating multiple elements on demand.

Here we demonstrate the deterministic transfer of optical-quality CNTs by utilizing anthracene as a sacrificial material. The use of large anthracene crystals allows transfer onto arbitrary substrates, and removal of anthracene through sublimation recovers the CNT luminescence intensity to the level of as-grown tubes. By monitoring the PL of CNTs during the transfer, we can select nanotubes of the desired chirality and position them with sub-micron spatial accuracy. As a demonstration of the chirality-on-demand transfer technique, we perform deterministic coupling to a photonic crystal nanobeam cavity. Furthermore, we design and assemble a cross junction from semiconducting CNTs to observe intertube exciton transport.

## Results and discussion
**Anthracene-assisted nanotube transfer**. Figure 1 illustrates the steps of the anthracene-assisted transfer of individual CNTs.

To pick up long CNTs and release them on arbitrary substrates with minimum contamination, we use thin and large-area organic crystals that do not require wet processes or high-temperature treatments. Anthracene, which consists of three benzene rings, meets such requirements[22,23]. We expect the similarity of molecular structure between anthracene and CNTs to lead to strong $\pi$–$\pi$ interactions[24], possibly improving the efficiency of picking up CNTs from the growth substrates. The charge-neutral nature of anthracene[25] should suppress exciton quenching, and we anticipate PL of CNTs to remain bright.

As shown in Fig. 1a, we have built a transfer system where the nanotube PL, the laser reflection, and the wide-field optical image are monitored during the nanotube transfer process. An anthracene single crystal is picked up with a glass-supported polydimethylsiloxane (PDMS) sheet (Gelfilm by Gelpak®)[26] (Fig. 1b). Viscoelastic nature of PDMS is utilized to tune the adhesion to anthracene crystals, broadening the choices of materials and shapes for nanotube growth substrates and receiving substrates. CNTs are picked up by pressing the anthracene/PDMS stamp against a substrate with as-grown CNTs (Fig. 1c), followed by quick separation (>10 mm/s) so that the anthracene crystal remains attached to the PDMS sheet[27]. PL mapping of CNTs on the anthracene crystal is performed when a CNT of a specific chirality needs to be selected (Fig. 1d). The stamp is then pressed on a receiving substrate. By slowly peeling off the PDMS (<0.2 μm/s), the anthracene crystal with the CNTs are released on the substrate (Fig. 1e). Sublimation of anthracene in air leaves behind clean CNTs on any substrate (Fig. 1f, g) because contamination from solvents is absent in the all-dry process. Figure 1h–k shows the optical microscopy images of anthracene crystals during the transfer steps, corresponding to the schematics in Fig. 1b, c, e, g, respectively. Being a single crystal, the anthracene stamp can be easily handled and transferred without significant damage even over trenches and pits, and can be sublimed after a typical heating process at 110 °C for 10 min.

**Clean and efficient transfer of CNTs from quartz**. We first apply the anthracene-assisted method for CNTs grown on a quartz substrate[19], where the CNTs grow parallel to each other with lengths exceeding 100 μm. We transfer the CNTs from the quartz substrate to trenches etched on a $SiO_2$/Si substrate. Figure 2a shows typical PL spectra from a transferred CNT with a total length of ~50 μm, which are measured at an air-suspended region over a 5-μm-wide trench (red) and at an on-substrate region (green). The PL efficiency of the suspended region is higher by ~250-fold than the on-substrate region of the same CNT. The absence of $SiO_2$ underneath also narrows and blue-shifts the emission spectra by fourfold and 55 meV, respectively. In Fig. 2b, a scanning electron microscopy (SEM) image displays uniformity along the entire length of the air-suspended CNT. In a PL image plotting the integrated PL intensity $I_{PL}$ over a ~50 nm spectral window centered at the emission wavelength $\lambda_{em}$ (Fig. 2c), we also observe the uniform optical response. The intratube variation of the $E_{11}$ peak positions (standard deviation of 2.3 nm) is smaller than the intertube variation in as-grown CNTs with the same chirality (standard deviation of 3.8 nm) reported in a previous study[28]. We assign the tube chirality to be (9, 8) from the PL excitation (PLE) map (Fig. 2d).

Insignificant residue on CNTs and the substrate surface is confirmed in atomic force microscopy (AFM) image of CNTs, and preservation of CNT alignment after the transfer is observed (Fig. 2e). The measured heights of the CNTs fall within the typical diameter range of those on the growth substrates[29]. It is noteworthy that the anthracene stamp gives rise to only one

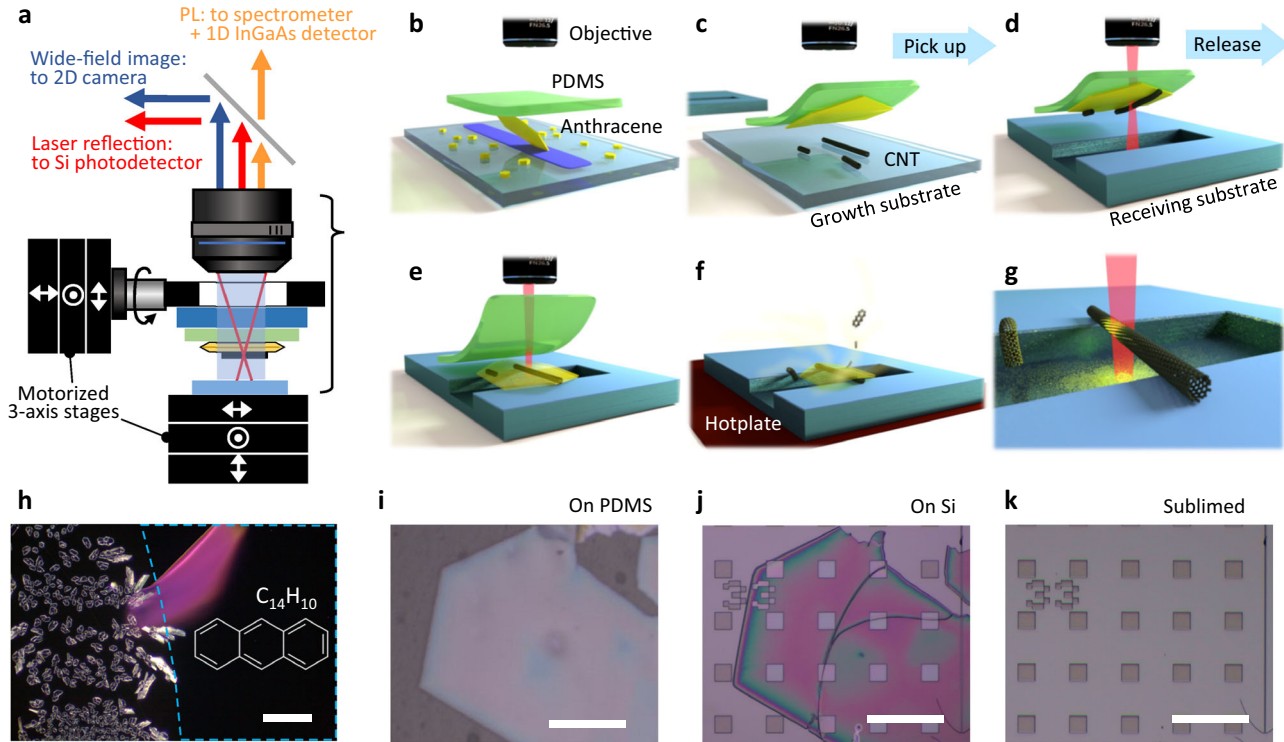

**Fig. 1 Procedure for anthracene-assisted dry transfer of carbon nanotubes. a** Schematic of the transfer system that simultaneously monitors the nanotube PL, the laser reflectivity, and the wide-field image. Substrates and a stamp are mounted on independent 3-axis stages. **b** Anthracene crystals are grown on glass slides through in-air sublimation and are picked up with a PDMS stamp. **c** CNTs are picked up with the PDMS/anthracene stamp, **d** followed by a PL measurement to locate CNTs of interest. **e** Anthracene/CNTs are released on a receiving substrate by peeling off the stamp. **f** In-air heating of the substrate removes the anthracene crystals and **g** leaves behind the CNTs alone. **h–k** Typical optical images of anthracene crystals grown on a glass slide (**h**), picked up on a PDMS (**i**), transferred to a Si chip with an array of pits (**j**), and after sublimation in air (**k**). Region outlined by a blue dashed box in **h** is coated with marker ink. Scale bars are 100 μm.

tenth as much contamination as a PDMS stamp (Fig. 2f), suggesting an advantage also for transferring 2D materials.

The use of anthracene as a medium drastically improves the overall transfer yield compared with the transfer using PDMS alone. Figure 2g shows an optical image of the anthracene crystal with CNTs underneath. G-band Raman mapping in Fig. 2h clearly indicates that CNTs are transferred only through anthracene mediation. The improved transfer yield of the anthracene-assisted method may partially originate from the fact that 100% of the picked-up CNTs are released on the receiving substrates together with the anthracene crystals. We should note that the pick-up efficiency by the anthracene crystals is much higher for small diameter CNTs (~1 nm), but the diameter range of CNTs on quartz under the current growth condition (1.42 ± 0.17 nm[29]) lies outside of the optimal range (Supplementary Fig. 3). Combined with the strong tube-substrate interaction that spontaneously aligns CNTs during the growth, the fraction of CNTs actually picked up is <1%. Optimization of the growth conditions should improve the pickup efficiency and further improve the transfer yield.

**High optical quality of the transferred CNTs**. The CNTs transferred over trenches should fluoresce as brightly as unprocessed ones in the absence of contamination. To evaluate the PL properties of the CNT shown in Fig. 2a, we perform PL measurements on typical substrates used for growth and photonic applications of CNTs; specifically, we use as-grown CNTs on quartz substrates[19] or over trenches, and transferred CNTs on poly(methyl methacrylate) (PMMA)[30] or polystyrene[31] (see the

Methods section). We measure >10,000 spectra on each sample to perform statistical characterization.

Figure 3a shows representative PL spectra of the brightest CNTs within each sample. As-grown CNTs on quartz show a broad peak with very small $I_{PL}$ due to the strong interaction with the substrate. The PL properties on the polymers are superior to those on quartz and $SiO_2$ (Supplementary Fig. 6) even in comparison with the previous study[32] likely due in part to the absence of wet processes and high-temperature treatments (see Supplementary Fig. 8). Not surprisingly, the as-grown suspended CNTs fluoresce most brightly and sharply out of all substrates studied.

The influence of each substrate observed in the representative spectra is further confirmed by statistical analysis. The majority of the measured spectra show only noise, implying each peak originates from at most one nanotube (see Supplementary Fig. 6). We therefore consider $I_{PL}$ above the noise floor to be representing the brightness of individual CNTs, whose histogram follows a log-normal distribution in all the samples reflecting the stochastic formation of exciton quenching sites (see Supplementary Fig. 7). When $I_{PL}$ at a relative frequency of $10^{-3}$ is used for comparison, the PL enhancement factor of the CNTs on the polymers with respect to those on quartz ranges between two to three orders of magnitude (the inset in Fig. 3a). Air-suspended CNTs have $I_{PL}$ greater by >4000-fold than unprocessed tubes on quartz. Importantly, the CNT shown in Fig. 2, whose $I_{PL}$ is indicated by the red vertical line in the inset of Fig. 3a, even outperforms the best as-grown suspended CNTs. It is somewhat surprising that PL recovery (~5000 fold) through the transfer from quartz surface to the trench results in intensity matching as-grown tubes.

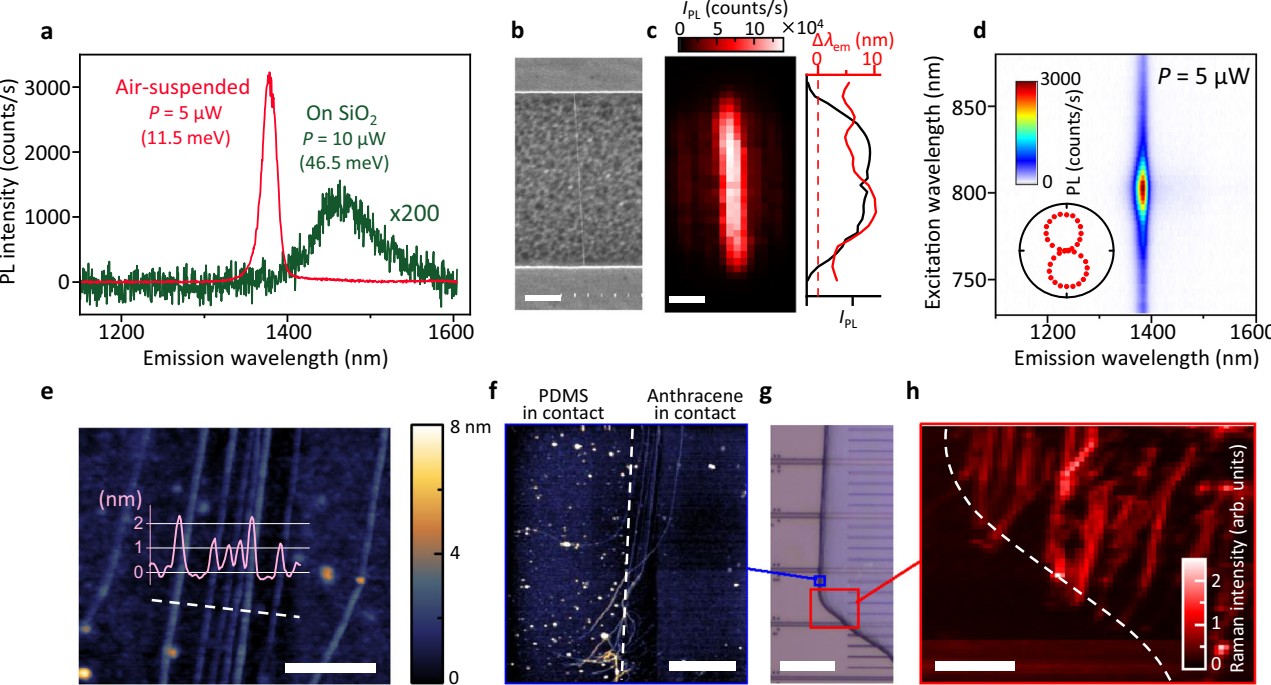

**Fig. 2 Bright PL of a transferred CNT. a** PL spectra of the CNT transferred from a quartz substrate. Red and green spectra are measured from the same CNT at an air-suspended region and at a region in contact with the $SiO_2$ surface, respectively. Numbers in parentheses represent the full width at half maximum. **b** SEM image of the suspended region and **c** the corresponding PL image at emission wavelength $\lambda_{em}$ of 1379 nm. Excitation power $P = 5 \mu W$. The peak shift $\Delta\lambda_{em}$ (red) from as-grown tubes[28] and integrated PL intensity $I_{PL}$ (black) along the tube axis are shown in the right panel. **d** PLE spectrum from which chirality is determined to be (9,8). Inset: polarization angle dependence of $I_{PL}$. **e, f** AFM image of the aligned CNTs measured on the $SiO_2$ surface after sublimation of an anthracene crystal. In **f**, the left and right sides are in contact with PDMS and anthracene, respectively. **g** Optical image of the anthracene crystal with CNTs from quartz. **h** Raman mapping showing G-band intensity in the region outlined by a red box in **g**. Scale bars in **b**–**c** and **e**–**h** are 1, 1, 0.5, 2, 50, and 10 μm, respectively.

We note that statistical evaluation for such transferred CNTs requires further improvement of nanotube pick-up efficiency through diameter optimization.

We also find that the bright PL is accompanied by narrow spectral linewidths with small statistical variance (see Supplementary Fig. 6). The air-suspended CNTs show the sharpest peaks, while CNTs on polymers have linewidths equivalent to those in aqueous agarose gels[33]. A slightly broadened linewidth is observed for the air-suspended CNT transferred from quartz in Fig. 2 compared with typical as-grown ones, which may indicate the presence of anthracene residue after the sublimation. The high PL intensity of the transferred CNT comparable to as-grown CNTs suggests that anthracene residue has a negligible effect on PL intensity.

If PL intensity also remains high for nanotubes on anthracene, chirality assignment can be done after picking up the CNTs. As the PL of CNTs on the growth substrate surface is barely detectable, the capability for assigning the tube chirality would be an important factor towards deterministic transfer. We perform PL measurements for the CNTs supported by anthracene crystals over the pits (Fig. 1j). Figure 3b shows a typical PLE map of a CNT on anthracene, peaking as sharply as the air-suspended ones at the $E_{11}$ and $E_{22}$ resonances. The tube orientation is revealed by both the PL image (Fig. 3c) and the polarization dependence of $I_{PL}$ (the inset). As shown in Fig. 3d, chiralities of such CNTs can be assigned from PLE spectra prior to releasing onto receiving substrates.

**Deterministic transfer under PL monitoring.** The 100% release of CNTs from the stamp to the target substrate combined with

the bright PL on anthracene facilitates the deterministic transfer of CNTs where chirality and position are controlled simultaneously. Position-controlled transfer approaches have been limited for CNTs[15,18] because individual CNTs are barely visible by conventional optical means. Electron microscopy can visualize CNTs, but it requires elaborate vacuum systems and induces exciton quenching sites[34]. Since PL spectroscopy can reveal the location, angle, and chirality of single CNTs, we have incorporated the CNT transfer system into the micro-PL setup. Hyperspectral 2D images and PLE maps are obtained to locate a CNT with a desired chirality among CNTs on anthracene crystal surface. As shown in Fig. 4a, we align the selected CNT and the destination on the receiving substrate to the optical axis of the laser beam. The anthracene crystal with the CNT is pressed and then released at the designated position.

We demonstrate such a deterministic approach by transferring the CNTs, whose chiral indices are pre-specified on a receiving substrate as shown in the upper part of Fig. 4b. A nanotube with chirality of (11,6) is found on the anthracene crystal and then transferred to the center of the cross mark. PL signals at $\lambda_{em} = 1418$ nm and the laser reflectivity are superimposed (Fig. 4b, bottom), displaying the desired CNT at the designated location with only ~500 nm misalignment. The PLE map confirms the tube chirality to be (11,6) (Fig. 4c), thereby demonstrating that our transfer method provides a sub-micron position accuracy and chirality selectivity for individual bright CNTs. We have subsequently transferred a (9,7) nanotube at the same location and the result is summarized in Supplementary Fig. 11, further supporting the reliability of our method.

The benefits of the position and chirality control over transferred CNTs are more evident in achieving deterministic

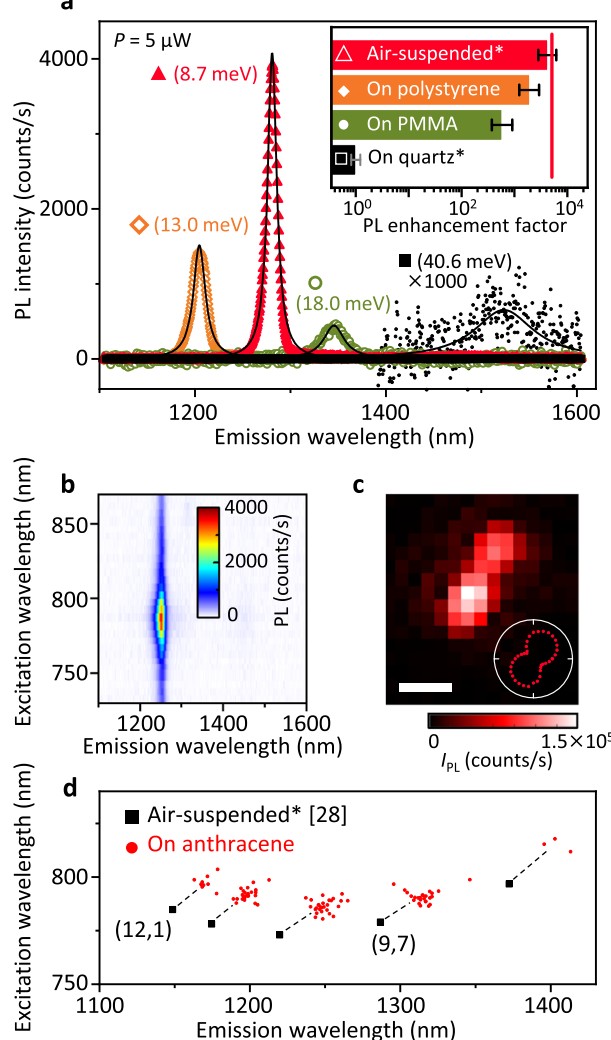

**Fig. 3 PL properties of CNTs on various surfaces. a** Typical PL spectra with top ~0.1% of integrated PL intensity $I_{PL}$ for each sample at excitation power $P = 5\,\mu W$. Black solid lines are Lorentzian fits. Numbers in parentheses represent the full width at half maximum. Inset: PL enhancement factor in comparison with CNTs on quartz, at $I_{PL}$ with a relative frequency of $10^{-3}$. Error bars are the $I_{PL}$ ranges that appear twice or half as frequently. Red vertical line represents the transferred air-suspended CNT in Fig. 2. The sample names with asterisks represent as-grown CNTs; otherwise, the CNTs are transferred via anthracene. **b** Typical PLE spectrum and **c** PL image at emission wavelength $\lambda_{em}$ of 1253 nm of a (10,5) CNT picked up on anthracene crystals. Inset: Polarization dependence of $I_{PL}$. Scale bar is 1 μm. **d** PLE peak positions of CNTs on anthracene (red dots), which are redshifted compared with as-grown suspended CNTs (black squares)[28]. $P = 100\,\mu W$ is used in **b**–**d**.

coupling of a CNT to a photonic crystal microcavity, in which amplified electric fields are localized within a few μm[35]. We demonstrate the capability of our transfer method by precisely placing a CNT whose emission wavelength matches the fundamental mode of a nanobeam cavity. As shown in Fig. 4d, a hexagonal boron nitride (h-BN) flake covers an as-fabricated nanobeam cavity and serves as a nano-spacer to simultaneously prevent exciton quenching and the decay of electric fields.

We perform deterministic transfer of a CNT and h-BN by taking into account the spectral shifts of the CNT and the cavity modes in contact with h-BN. The fabrication steps are summarized in Fig. 4e. The fundamental mode of a test nanobeam cavity is redshifted by 27.6 nm after the transfer of an h-BN flake (Fig. 4f, top). Based on the shift amount, we seek a target nanobeam whose cavity mode under this particular h-BN flake would match the (13,5) tube emission on h-BN[36] (Fig. 4f, middle). After characterizing a (13,5) CNT on an anthracene crystal, we further pick up the h-BN flake and place this anthracene/CNT/h-BN heterostructure onto the target nano-beam, followed by anthracene sublimation. A PL spectrum of the CNT in Fig. 4g has a cavity-coupled narrow line, exhibiting a nearly perfect spectral and spatial matching. We note that the high compatibility of anthracene crystals with 2D materials plays a key role in this demonstration (see Supplementary Fig. 9 for the case of suspended monolayer WSe$_2$).

**Exciton transfer at a cross junction.** Our transfer technique also offers opportunities to study basic elements needed to build advanced structures consisting of multiple CNTs. Cross junction formation is an example, which allows us to extract intertube exciton dynamics at controlled interfaces. A PL image and a PLE map of an isolated (10,5) tube that is transferred first are shown in Supplementary Fig. 12. Next, we find a (13,5) tube among the CNTs picked up from the quartz substrate, whose angle is adjusted perpendicular to the pre-transferred (10,5) tube. The (13,5) tube and the anthracene crystal are then released over the (10,5) nanotube on the chip.

We perform PL imaging with two different excitation conditions. Figure 5a shows $I_{PL}$ from the (13,5) tube when the excitation energy and polarization are resonant and parallel, respectively, showing a straight geometry of the transferred CNT. When the laser wavelength and polarization are chosen to maximize the $I_{PL}$ of the (10,5) tube, the PL image for the (10,5) tube emission does not show any remarkable change before and after the transfer of the additional tube (Supplementary Fig. 12a and Fig. 5b). Interestingly, the PL image filtered around the (13,5) tube emission peak (Fig. 5c) shows a small bump that spatially overlaps with the (10,5) tube. This implies that higher energy excitons generated in the thinner (10,5) tube diffuse towards the cross junction before transferring into the other nanotube.

We also observe spectral signatures of intertube exciton transfer in PLE spectra. The PLE map in Fig. 5d is measured at the yellow star in Fig. 5a, showing the $E_{22}$ absorption peak of the (13,5) tube without other prominent absorption peaks. In contrast, when tubes are excited at the light blue diamond in Fig. 5b, c, the PLE spectrum for the (13,5) tube emission shows a sub-peak at the $E_{22}$ absorption peak of the (10,5) tube (Fig. 5e). This absorption peak for the (13,5) tube emission indicates that a small portion of excitons generated in the (10,5) tube is transferred and contributes to the emission around 1500 nm (Fig. 5f). The excitonic energy mismatch between the CNTs, as well as the 0D-like ultrasmall contact, may lead to the low efficiency of exciton transfer of ~8% (see Supplementary Fig. 12) compared with a previous study[37].

For more complex structures, the throughput of our deterministic transfer technique will need to be optimized. Room for technical improvement may lie in enhancing the tube-anthracene interaction (see Supplementary Fig. 3) and faster chirality assignment methods[38] compatible with our transfer process. With higher pick-up efficiency for nanotubes of known chirality, multiple components can be assembled on a single anthracene crystal in a similar manner to the cases in the stacking of 2D materials[39] before being transferred on a receiving substrate. The entire transfer process could then be even faster than polymer-based techniques, as temperature switching[40] is not required between the pick-up and release steps.

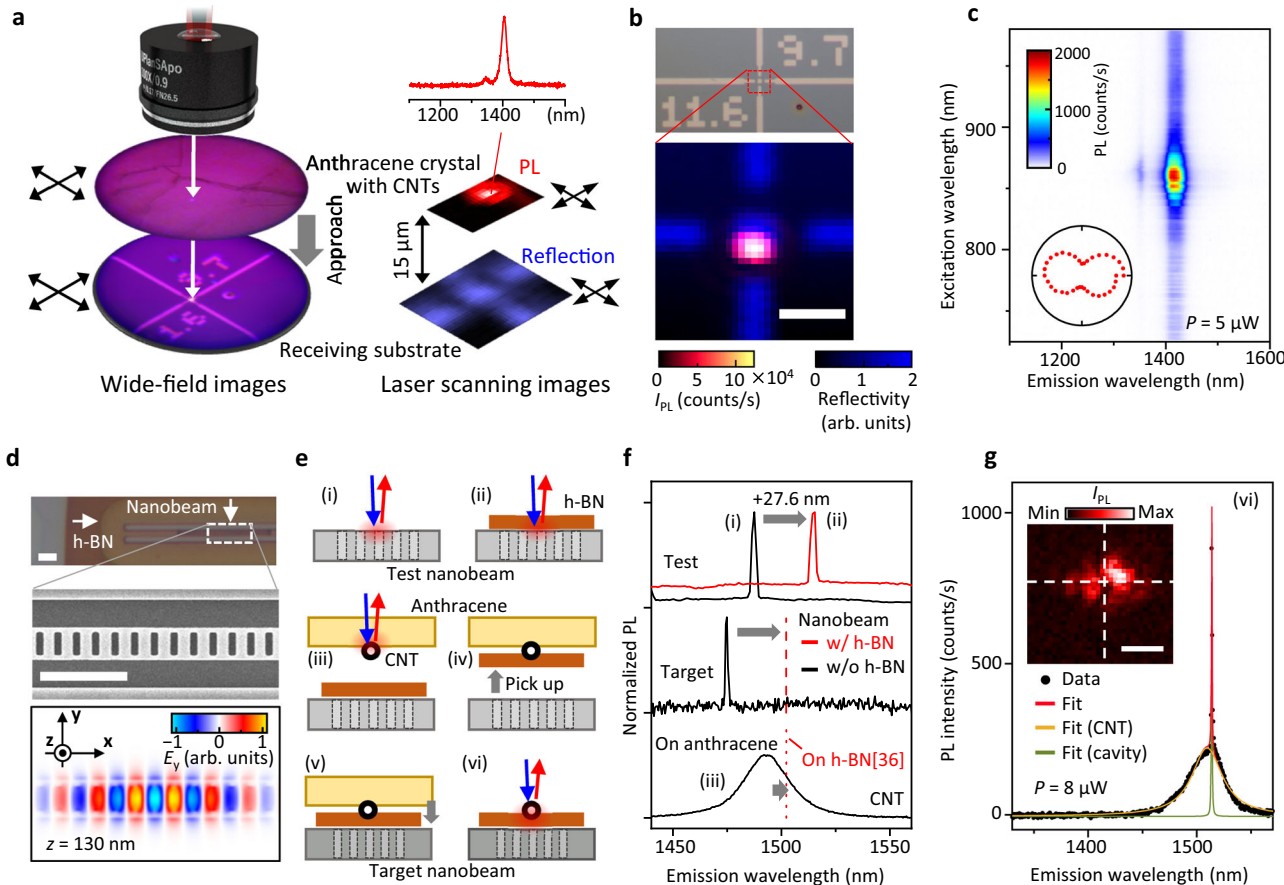

**Fig. 4 Demonstration of deterministic transfer of CNTs. a** Schematic showing the alignment of a selected CNT and a receiving substrate. Wide-field images and laser scanning images are simultaneously obtained for the stamp and the substrate. **b** Optical images of the receiving substrate, which is covered by a polystyrene film to suppress exciton quenching (see Fig. 3). Top panel shows a $35 \times 65 \, \mu m^2$ micrograph, in which the numbers (11,6) are lithographically patterned metal, and represent the chiral index of a CNT to be transferred. Confocal PL image of the transferred (11,6) nanotube (bottom), where integrated PL intensity $I_{PL}$ at 1418 nm and the laser reflectivity are superimposed. **c** PLE map of the transferred CNT. Inset: Polarization dependence of $I_{PL}$. Data in **b** and **c** are taken with excitation power $P = 5 \, \mu W$. **d** Optical micrograph of a nanobeam cavity covered with a ~30-nm-thick h-BN flake. Typical SEM image of an as-fabricated nanobeam cavity (middle) and simulated spatial distribution of the $y$-component of the electric field for the fundamental mode at the top surface (bottom). **e** Schematics showing the steps for measurements and deterministic transfer of a CNT/h-BN heterostructure onto a nanobeam cavity. **f** PL spectra of the fundamental mode of cavities with (red) and without (black) the h-BN nano-spacer, and PL spectrum of a (13,5) nanotube picked up on an anthracene crystal (bottom). For the test nanobeam (top), the resonant wavelength is redshifted by 27.6 nm due to the covering dielectric layer. Based on the shift value, a target nanobeam (middle) is chosen so that wavelengths of the shifted fundamental mode and the (13,5) tube emission[36] coincide at the red dashed (dotted) line indicating the expected peak position under (on) the h-BN. **g** PL spectrum of the cavity-coupled CNT. Black dots are data and solid lines are the Lorentzian multi-peak fits. $P = 8 \, \mu W$ and laser polarization is perpendicular to the nanobeam. Inset: PL image of the (13,5) tube emission. White dashed cross indicates the center of the cavity. All the scale bars are 2 μm.

In conclusion, we have developed a method for the deterministic transfer of chirality-on-demand CNTs with high optical quality, assisted by sacrificial anthracene single crystals. Transferred CNTs fluoresce as bright as unprocessed air-suspended CNTs, where crystal growth and removal by sublimation play a key role. With precise control over position, angle, and tube chirality, individual nanotubes are coupled in a deterministic manner to other nanoscale components such as a nanobeam cavity and another nanotube. Our results demonstrate the extensive capabilities of the deterministic transfer method in nanodevice fabrication from atomically defined building blocks. Novel materials and automated fabrication systems would allow for larger scale and higher complexity structures, eventually leading to technology that makes use of all the room at the bottom.

## Methods

**Micro-PL measurements**. A homebuilt confocal microscopy system is used to perform PL measurements at room temperature in air. We use a wavelength-tunable Ti:sapphire laser for excitation with its power and polarization angle controlled by neutral density filters and a half-wave plate, respectively. The laser beam is focused on the samples using an objective lens with a correction collar, a numerical aperture of 0.65, and a working distance of 4.5 mm. The $1/e^2$ beam diameter and the collection spot size defined by a confocal pinhole are ~1.2 and 5.4 μm, respectively. PL is collected through the same objective lens and detected using a liquid-nitrogen-cooled InGaAs diode array attached to a spectrometer. To assist the transfer process while monitoring PL signals, wide-field 2D images of the substrate and the stamp are simultaneously obtained with a commercially available CMOS camera under coaxial illumination by an LED (see Supplementary Fig. 1).

**CNT synthesis**. We use three types of CNT synthesis in this study. The arrays of long CNTs are grown on single-crystalline quartz substrates[29]. Iron catalyst nanoparticles with a nominal thickness of 0.2 nm are deposited on lithographically patterned stripes. The CNT arrays are grown at 800 °C for 10 min by low-pressure alcoholic chemical vapor deposition (CVD) technique. Air-suspended CNTs are grown across 1- to 2-μm-wide trenches that are formed on $SiO_2$/Si substrates by dry etching. Iron catalyst (0.1-nm thick) is deposited near the trenches by electron-beam evaporation, followed by atmospheric pressure CVD (1 min) at 800 °C with ethanol precursor supplied through bubbling of $Ar/H_2$ carrier gas[28]. For randomly oriented CNTs that are transferred via anthracene, the CNTs are grown on flat

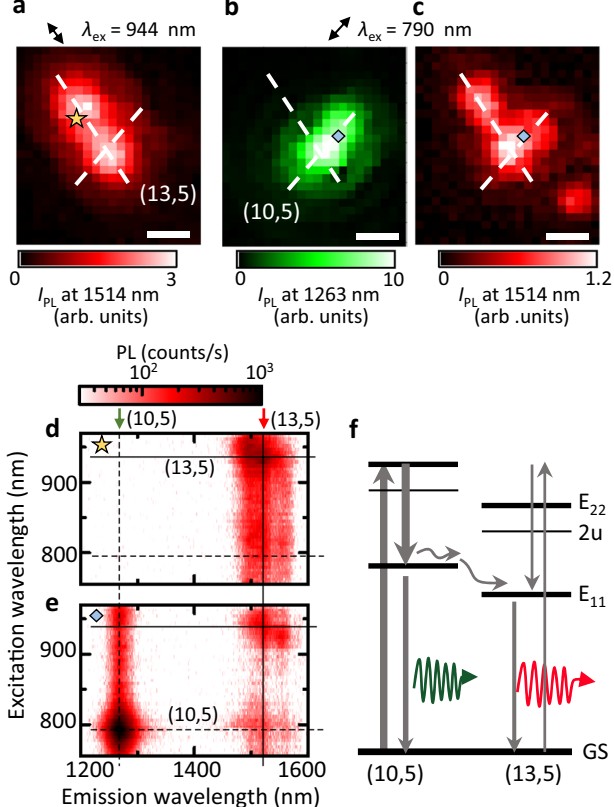

**Fig. 5 Crossed CNTs formed through multiple transfer processes. a** Confocal PL image of a transferred (13,5) CNT, which has an emission peak at 1514 nm, with the excitation at its $E_{22}$ resonance (944 nm) and the laser polarization parallel to the tube axis. **b** PL image of a (10,5) tube at emission wavelength $\lambda_{em}$ of 1263 nm, taken with excitation near its $E_{22}$ resonance (790 nm) and the polarization parallel to its axis. **c** PL image at $\lambda_{em} = 1514$ nm, which is simultaneously obtained with **b**. Scale bars are 1 μm. **d, e** PLE maps measured at the yellow star and the blue diamonds in **a**–**c**. **f** Schematic of energy diagram for the (10,5)–(13,5) CNT heterostructure, showing emission at 1263 and 1514 nm with excitation wavelength $\lambda_{ex} = 790$ nm. $P = 100$ μW for all the panels. The substrate is covered with a PMMA film before the CNT transfer, and measurements are performed without removing the anthracene crystal used for transferring the (13,5) tube.

$SiO_2/Si$ substrates without trench formation nor catalyst patterning, and the other growth conditions are the same as the air-suspended CNTs.

**Anthracene crystal growth**. Anthracene crystals are grown by in-air sublimation[41]. Anthracene powder is placed on a glass slide that is heated to 80 °C, and another glass slide is placed above the anthracene source typically with a 1 mm spacing. Thin and large-area single crystals start growing on the glass surface but extend out of plane of the glass surface. To efficiently grow such thin and large-area crystals, glass slides are patterned with ink of commercially available markers, on which the nucleation of three-dimensional crystals is suppressed. The typical growth time is 10 h.

**Transferred CNT samples for PL comparison**. In Fig. 3a, transferred CNTs (shown without asterisk) are prepared using the anthracene-assisted method. Polymer (PMMA and polystyrene) thin films are formed by spin-coating on Si chips with alignment markers, followed by baking at 170 °C. CNTs on anthracene are measured at suspended parts of anthracene/CNTs over square pits on $SiO_2/Si$ chips (see Fig. 1j). All the transferred samples measured in Fig. 3 and Supplementary Fig. 6 are grown from unpatterned catalysts on flat $SiO_2/Si$ substrates. For air-suspended CNTs, we scan the excitation laser beam along trenches with excitation wavelength $\lambda_{ex} = 780$ nm, while 2D raster scan with excitation at $\lambda_{ex} = 790$ nm is performed for the rest of the samples to adjust for the screening-induced energy shifts[42]. We acquire the PL signal of CNTs on quartz without a grating to

increase the signal-to-noise ratio and calibrate the intensity by taking into account the grating efficiency to compare $I_{PL}$ with other samples.

**Deterministic coupling with nanobeam cavities**. Air-mode nanobeam cavities are fabricated using silicon-on-insulator wafers with a top Si thickness of 260 nm and a buried oxide thickness of 1 μm. The nanobeams are patterned by electron beam lithography, followed by inductively coupled plasma etching of the top Si layer. The buried oxide layer under the nanobeam structures is then etched by hydrofluoric acid. To align the wavelengths of a transferred CNT and a nanobeam cavity with h-BN in between, cavity modes of the nanobeams are measured using Si PL with $\lambda_{ex} = 780$ nm and $P = 200$ μW.

## Data availability
The data that support the plots within this paper and other findings of this study are available from the corresponding authors upon reasonable request.

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

## Acknowledgements

Parts of this study are supported by JSPS (KAKENHI JP20H02558, JP19J00894, JP20K15137), MIC (SCOPE 191503001), and MEXT (Nanotechnology Platform JPMXP09F19UT0075). The growth of h-BN crystals is supported by the Element Strategy Initiative conducted by the MEXT (JPMXP0112101001), JSPS (KAKENHI JP20H00354), and JST (CREST JPMJCR15F3). K.O. and D.Y. are supported by JSPS (Research Fellowship for Young Scientists). N.F. is supported by RIKEN Special Post-doctoral Researcher Program. We thank T. Inoue, S. Maruyama, and K. Nagashio for help in the preparation of CNTs and 2D materials, as well as the Advanced Manufacturing Support Team at RIKEN for technical assistance.

## Author contributions

K.O. developed the nanotube transfer system, and performed all the nanotube-related experiments and data analysis. N.F. performed transfer and spectroscopy of 2D materials. D.Y. performed simulation, fabrication, and measurements of nanobeam cavities. T.T. and K.W. provided boron nitride crystals. Y.K.K. supervised the project. K.O. and Y.K.K. co-wrote the manuscript. All the authors commented on the manuscript.

## Competing interests

The authors declare no competing interests.
