## [Peer Review File · Nature Communications]

REVIEWER COMMENTS

Reviewer #1 (Remarks to the Author):

In this work K. Otsuka and collaborators report on the deterministic transfer of individual single wall carbon nanotubes on various devices of technological interest with full control over the chiral species, the position and the orientation of the nanotube. This new technique is based on the combination of a dry transfer approach using a sacrificial anthracene layer and of the real-time optical monitoring of the nanotube properties. In particular, photoluminescence is used for its ability to provide accurate determination of the chiral species together with a good assessment of the tube cleanliness.

The potential of this approach is exemplified with two realizations. First, the author demonstrate the deterministic coupling of a nanotube to a nano-beam photonic crystal with full spectral and spatial matching for PL enhancement through the Purcell effect. Next, they build a cross-junction between two semi-conducting nanotubes to study the inter-tube exciton transfer processes. By the way, the authors show that the method is also applicable for 2D materials like transition metal dichalcogenides which broadens further the scope of the technique.

This technique represents a significant breakthrough in nanotechnology and makes the dream of the early 2000s' come true : to put on demand a single nanotube of determined chirality at an accurate location, with minimal pollution of the sample. Nevertheless, the issue of the throughput of this approach remains to be addressed. It would also be interesting to check the robustness of the method regarding the electronic devices, especially at low temperature.

Technically, the main paper and the methods are sound and well documented. The supplementary section brings clear and relevant information to the reader.

In view of this significant and inspiring technological achievement, I suggest to publish this work in Nature Communications when the authors have addressed the following minor comments/suggestions :

- line 27 missing word "of" in the citation
- figure 4d : please specify the orientation of the axis x, y and z
- figure 4e : what is the purpose of steps iv ? why not directly stamp the nanotube on the hBN layer at the right location ?
- figure 4f : the description of the middle trace is not clear
- line 217 : the author claim a nearly perfect spatial matching. How to prove this statement with an accuracy of "only" 500nm ? What is the size of the optical mode ?
- regarding the cross-junction and exciton transfer (figure S12d and main text line 234 and next) : why does the emission at 1514nm (supposedly from the (13,5) nanotube) show up on the right side of the (13,5) nanotube, that is rather on the (10,5) nanotube than on the (13,5) nanotube ? In addition, the protocol to subtract accurately the non resonantly excited PL from the (13,5) nanotube is not very clear (SI). I would encourage the author to elaborate a little bit about this point. Same for the assessment of the transfer efficiency.
- SI section S6 : could the author comment on a possible correlation between the brightness of the nanotube and the spectral width (and show the corresponding data) ? In case such correlation exists, could this be addressed with the random defect model by generating pure dephasing on each defect on top of quenching ?
- section S7 : The authors use a diffusion length of 700 nm without any further comment. How does the value of this diffusion length change the results of their model ? More

importantly, equation S4 is unclear to me : what do the parameter α_i and λ_i stand for ? What is the physical meaning of this alternative model ?

Reviewer #2 (Remarks to the Author):

The manuscript "Deterministic transfer of optical-quality carbon nanotubes for atomically defined technology" by K. Otsuka et. al. is a very interesting study that uses anthracene, a polyaromatic hydrocarbon that sublimates at $\sim 300^\circ\text{C}$, to transfer individual single-wall carbon nanotubes (SWNTs) on a desired location of a substrate. The work is impressive and the demonstrated capability to transfer SWNTs with a remarkable precision can open a myriad of opportunities to study this 2D carbon material, which continues to provide excitement. Further, the demonstrated technique advances our abilities to fabricate devices with atomic precision. Importantly, the transferred SWNTs are left in a pristine state. The authors demonstrate the potential of their technique by optical measurements and show bright PL of the transferred SWNTs. They even found a small exciton transfer between two crossed carbon nanotubes. It is very impressive that they can achieve this cross structure. The authors use as an evidence for exciton transfer, the appearance of a peak at 1500 nm (assigned to (13,5) nanotube) when the excitation light is positioned at (10,5) SWNTs. A stronger evidence of exciton transfer would be PL lifetime measurement of (10,5) nanotube before and after adding the (13,5) SWNT on top.

The study also demonstrates transfer of MoS_2 using the anthracene-assisted technique.

The authors mention that the transfer yield is relatively low (do not consider this as a criticism, the reviewer appreciates the rational argument that the yield can be improved with process optimization). I was questioning if the presence of defects in the nanotube affects the yield? Taking into account the benzenoid-structure of the anthracene, I anticipate that the pi-pi interactions with the nanotube walls will be affected by the presence of defects. Thus, the technique is applicable (selective) to perfect SWNTs?

I strongly recommend the manuscript for publication.

I have some comments, driven mainly by the fact that adding some experimental details will make this technique more popular and easier to reproduce.

1. How is the PDMS peeled off without picking up the anthracene in the last step (row 86 -91 in the main text)? In the text it is claimed that fast peeling can pick up the anthracene and slow peeling only picks up the PDMS. How do you define fast peeling?
2. Is this technique universal for all kinds of substrates? In this study the authors transferred anthracene/SWNT from glass to Si substrate. What about glass to glass or glass to a more-hydrophobic surface?
3. About the anthracene growing technique (280-286). The substrate is first patterned with marker ink. Did they just draw a pattern on the substrate? And what kind of marker is used, water based/alcohol based? What is the color of the marker? Why does the ink suppress the nucleation of anthracene? A more detailed explanation on the procedure and mechanism will be helpful to reproduce the experiment.

Response to reviewer report for manuscript NCOMMS-20-48650-T by Otsuka *et al.*

We thank the reviewers for taking their time to review the manuscript, and we are delighted that both reviewers have found our results to be significant or impressive. We also thank the reviewers for their helpful comments, and we have used them to improve our manuscript. Our point-by-point response is provided below, followed by a summary of changes.

Response to Reviewer #1:

In this work K. Otsuka and collaborators report on the deterministic transfer of individual single wall carbon nanotubes on various devices of technological interest with full control over the chiral species, the position and the orientation of the nanotube. This new technique is based on the combination of a dry transfer approach using a sacrificial anthracene layer and of the real-time optical monitoring of the nanotube properties. In particular, photoluminescence is used for its ability to provide accurate determination of the chiral species together with a good assessment of the tube cleanliness.

We thank the reviewer for concisely summarizing our approach for the deterministic and clean transfer of carbon nanotubes.

The potential of this approach is exemplified with two realizations. First, the author demonstrate the deterministic coupling of a nanotube to a nano-beam photonic crystal with full spectral and spatial matching for PL enhancement through the Purcell effect. Next, they build a cross-junction between two semi-conducting nanotubes to study the inter-tube exciton transfer processes. By the way, the authors show that the method is also applicable for 2D materials like transition metal dichalcogenides which broadens further the scope of the technique.

We again thank the reviewer for summarizing our key realizations, and for commenting on the further applicability of our technique to 2D materials.

This technique represents a significant breakthrough in nanotechnology and makes the dream of the early 2000s' come true : to put on demand a single nanotube of determined chirality at an accurate location, with minimal pollution of the sample. Nevertheless, the issue of the throughput of this approach remains to be addressed. It would also be interesting to check the robustness of the method regarding the electronic devices, especially at low temperature.

We are very glad to hear that the reviewer notes our technique as “a significant breakthrough in nanotechnology”. We do recognize the throughput of our approach could be an issue when one seriously tries to build devices out of a lot of nanomaterials. We should emphasize, however, that stacking of different nanotubes/materials can be repeated on a single anthracene crystal before the stacked materials are released on a target substrate in a similar manner to previous studies (such as Masubuchi *et al.*, *Nat. Commun.* **9**, 1413 (2018)). In our process, one can even perform another transfer process using the same PDMS stamp but with a new anthracene crystal without formation of a sacrificial polymer layer that requires solvent evaporation. The throughput in this study will be then limited by the mounting of the stamp, the searching of desired CNTs, and the

alignment to the target, but time required for those steps can be significantly reduced by automated mechanical systems, more sophisticated software, and real-time imaging using a 2D InGaAs array. We have added a paragraph that comments on the throughput of our technique (line 250-258). We agree that it would be interesting to check our method for electronic devices, as we expect them to be more sensitive to contamination at very low temperatures.

Technically, the main paper and the methods are sound and well documented. The supplementary section brings clear and relevant information to the reader.

We thank the reviewer for mentioning that our manuscript, methods, and supplementary section are well documented.

In view of this significant and inspiring technological achievement, I suggest to publish this work in Nature Communications when the authors have addressed the following minor comments/suggestions

We are very happy to hear that the reviewer suggests the publication of our manuscript in Nature Communications with an evaluation of our work as “*significant and inspiring technological achievement*”. We have addressed the questions below.

- line 27 missing word “of” in the citation

We thank the reviewer for correcting our careless mistake in citing such a famous phrase.

- figure 4d : please specify the orientation of the axis x, y and z

We appreciate the reviewer’s comment, and have defined the orientation of the axes x, y, and z in Figure 4d.

- figure 4e : what is the purpose of steps iv ? why not directly stamp the nanotube on the hBN layer at the right location ?

We thank the reviewer for bringing up this point. A brief answer to this question is because the nanotube emission wavelengths take discrete values depending on the chirality. To get good spectral overlap, we need to find a cavity mode that matches the nanotube emission after placing an h-BN flake, but the exact shift caused by h-BN cannot be predicted. For example, we found the cavity mode under the h-BN flake (red line in Figure 4f) peaked at a slightly longer wavelength than the (13,5) nanotube emission on h-BN flakes (red dotted line in Figure 4f), according to our recent study (Fang *et al.*, *ACS Photonics* **7**, 1773 (2020)). For better spectral coupling, we picked up the h-BN flake (step iv) and sought another nanobeam cavity that would match the nanotube emission in the presence of this particular h-BN flake (step v). It is worth mentioning that picking up such a thin h-BN flake on SiO₂/Si surface cannot be done with a conventional PDMS stamp, but is made possible by anthracene layers due to the very flat nature of the single-crystalline anthracene.

- figure 4f : the description of the middle trace is not clear

We thank the reviewer for pointing out the inadequacy of our description. Related to the question above (and our response), the middle trace in Figure 4f shows the raw cavity mode of a nanobeam which was later coupled to the nanotube emission to show the spectra in Figure 4g. We have added a description on this point in the figure caption.

- line 217 : the author claim a nearly perfect spatial matching. How to prove this statement with an accuracy of “only” 500nm ? What is the size of the optical mode ?

The envelope of E_y^2 (square of electric field of y-component) has the 1/e full width of $\sim 2.5 \mu\text{m}$ (calculated from the data in Figure 4d), and the spatial misalignment of the cavity center (cross in the inset of Figure 4g) and the transferred CNT (brightest spot in the PL mapping) is $< 1 \mu\text{m}$. Considering the spatial tolerance of our nanobeam cavity, we regard the spatial matching in the experiment as “nearly perfect”.

- regarding the cross-junction and exciton transfer (figure S12d and main text line 234 and next) : why does the emission at 1514nm (supposedly from the (13,5) nanotube) show up on the right side of the (13,5) nanotube, that is rather on the (10,5) nanotube than on the (13,5) nanotube ? In addition, the protocol to subtract accurately the non resonantly excited PL from the (13,5) nanotube is not very clear (SI). I would encourage the author to elaborate a little bit about this point. Same for the assessment of the transfer efficiency.

In our confocal micro-PL system, the $1/e^2$ laser beam diameter is $\sim 1.2 \mu\text{m}$, which is smaller than the collection spot size defined by a confocal pinhole of $5.4 \mu\text{m}$. We therefore measure excitation images, instead of emission images (or wide-field images). In the present case, when the (10,5) nanotube is excited by the laser at the blue diamond (in Figure 5c), the emission from the (13,5) nanotube ($\sim 1514 \text{ nm}$) one micron away from the laser beam was detected. We have added information on the laser beam size and the collection spot size in Methods section.

We also thank the reviewer for pointing out our inadequate explanation on the protocol for subtracting non-resonantly excited PL from the (13,5) tube. We have added several sentences and equations on this point in the supplementary information.

- SI section S6 : could the author comment on a possible correlation between the brightness of the nanotube and the spectral width (and show the corresponding data) ? In case such correlation exists, could this be addressed with the random defect model by generating pure dephasing on each defect on top of quenching ?

We appreciate the reviewer’s interest in the brightness and the spectral width of nanotube emission. We have compared these properties within the same type of samples and found no clear correlation (please see the newly added Figure S6e). In addition, the CNT on PMMA and anthracene have a large variation of emission wavelength within a single tube due to inhomogeneous interaction with the support, while as-grown suspended CNTs do not show such an inhomogeneous emission. Therefore, we suspect the spectral width variation does not directly

originate from the defect-induced dephasing, though hidden links between the quenching defects and the inhomogeneity could exist. We have added a comment on this point in the supplementary information S6.

- section S7 : The authors use a diffusion length of 700 nm without any further comment. How does the value of this diffusion length change the results of their model ? More importantly, equation S4 is unclear to me : what do the parameter α_i and λ_i stand for ? What is the physical meaning of this alternative model ?

We adopt the diffusion length of 700 nm by referring to values in Fig 5b of Ref. [S4] (Ishii *et al.*, *PRB* **91**, 125427 (2015)). As the values depend on tube chirality, we use a representative value. We have added a reference number in the supplementary information.

We also thank the reviewer for suggesting us to consider the dependence of our model for different diffusion lengths. As suggested, more Monte Carlo simulations are performed (Figure S7e,f), where two slopes are found in the λ -dependent emission intensity (note that it is shown in log-scale for clarity). The presence of the two slopes for all different diffusion lengths is the premise of our biexponential model (equation S4). We would like to remind that our model is rather phenomenological, which somehow reproduces the I_{PL} dependence on the number of quenching sites λ and in turn the experimental I_{PL} distribution. In Figure S7e,f, the I_{PL} vs λ can be fitted by biexponential curves, but the single exponential function also fits the dependence if the effective diffusion length is short (including the range of our experiment). Although we have not found an analytical derivation of our model yet, we believe the experimental I_{PL} distribution can be attributed to stochastic formation of quenching sites considering this phenomenological model.

Response to Reviewer #2:

The manuscript “Deterministic transfer of optical-quality carbon nanotubes for atomically defined technology” by K. Otsuka et. al. is a very interesting study that uses anthracene, a polyaromatic hydrocarbon that sublimates at $\sim 30^\circ\text{C}$, to transfer individual single-wall carbon nanotubes (SWNTs) on a desired location of a substrate. The work is impressive and the demonstrated capability to transfer SWNTs with a remarkable precision can open a myriad of opportunities to study this 2D carbon material, which continues to provide excitement. Further, the demonstrated technique advances our abilities to fabricate devices with atomic precision. Importantly, the transferred SWNTs are left in a pristine state. The authors demonstrate the potential of their technique by optical measurements and show bright PL of the transferred SWNTs. They even found a small exciton transfer between two crossed carbon nanotubes. It is very impressive that they can achieve this cross structure.

We sincerely thank the reviewer for concisely describing the results.

The authors use as an evidence for exciton transfer, the appearance of a peak at 1500 nm (assigned to (13,5) nanotube) when the excitation light is positioned at (10,5) SWNTs. A stronger evidence of exciton transfer would be PL lifetime measurement of (10,5) nanotube before and after adding the (13,5) SWNT on top.

We thank the reviewer for suggesting PL lifetime measurements as a stronger evidence for exciton transfer. We would like to observe exciton dynamics at nanotube cross junctions with various combination of chirality (n,m) or inter-tube separation using h-BN as a ultrathin spacer, as well as other forms of energy (phonon, electron, etc.).

The study also demonstrates transfer of MoS₂ using the anthracene-assisted technique.

We thank the reviewer for picking up the capability of our technique in 2D materials, as we hope it will also be implemented in 2D materials research fields.

The authors mention that the transfer yield is relatively low (do not consider this as a criticism, the reviewer appreciates the rational argument that the yield can be improved with process optimization). I was questioning if the presence of defects in the nanotube affects the yield? Taking into account the benzenoid-structure of the anthracene, I anticipate that the pi-pi interactions with the nanotube walls will be affected by the presence of defects. Thus, the techniques is applicable (selective) to perfect SWNTs?

We thank the reviewer for kindly understanding our comment on the low transfer yield, and also for bringing up the interesting point of defects. We have checked D/G ratio in Raman spectra for as-grown CNTs on quartz, and those transferred on SiO₂/Si substrate via PMMA or anthracene (Figure S10g-i has been added). The data show a slight increase of D/G ratio after transfer via anthracene, whereas the D/G ratio increase by PMMA transfer is less significant. Thus, we cannot claim the selectivity for perfect CNTs.

Instead, we would like to mention another clue for improved yield of the transfer than the diameter dependence. When we transferred randomly oriented CNTs grown on a SiO₂/Si substrate to another substrate via an anthracene crystal, the nanotube orientation had three preferred angles, which are nearly parallel to the edges of hexagonal anthracene crystals (please see the figure below). Just like van der Waals interaction between CNTs and single-crystalline quartz has preferred angles, the anthracene crystal also should have such a preference. Therefore, orientation matching between CNTs and anthracene crystals might improve the transfer yield or can be utilized for the control of nanotube orientation.

Figure: (a) Optical image of an anthracene crystal used for the transfer of CNTs. Arrows represent the angles of characteristic edges of the crystal. (b-d) Histograms of tube orientation obtained by polarization-dependent PL measurements for all measured nanotubes (b), (11,3) tubes (c), and (10,5) tubes (d). CNTs are originally grown on an SiO₂/Si substrate and does not have any preferred angle at the growth stage.

I strongly recommend the manuscript for publication.

We are very happy to hear that the reviewer strongly recommends the manuscript for publication.

I have some comments, driven mainly by the fact that adding some experimental details will make this technique more popular and easier to reproduce.

We sincerely thank the reviewer for providing us insightful comments. We have addressed the questions below.

1. How is the PDMS peeled off without picking up the anthracene in the last step (row 86 -91 in the main text)? In the text it is claimed that fast peeling can pick up the anthracene and slow peeling only picks up the PDMS. How do you define fast peeling?

It is widely known that the PDMS stamp technique can print a wide variety of objects onto any smooth substrates due to widely tunable adhesion through separation speed (Ref. [27] Meitl et al.,

Nat. Mater. **5**, 33 (2006)). In the last step (row 86-91), slow peeling reduces only the adhesion between PDMS and anthracene, but not that between anthracene and the substrate. For better control over the separation (peeling-off) speed and the spatial alignment, we used motorized stages both for the sample-mounting stage and stamp-mounting stage. Typical separation speed for picking-up (releasing) objects is >10 mm/s (<0.2 $\mu\text{m/s}$) as described in the manuscript (lines 89 and 92). To make this point clear, we have added a sentence on line 85-87.

2. Is this technique universal for all kinds of substrates? In this study the authors transferred anthracene/SWNT from glass to Si substrate. What about glass to glass or glass to a more-hydrophobic surface?

As we also mentioned the above response, the adhesion of PDMS to anthracene crystals can be tuned in a wide range. It is fairly easy to release anthracene crystals (CNTs) on glass or quartz substrates. We should also mention that our collaborator successfully transferred anthracene crystals onto Mo-based TEM grids, which have rough surface and small contact area with the crystals, by applying shear force to the PDMS stamp. We have added a more explicit comment on the wide range of options for substrates in the main text (lines 85-87).

3. About the anthracene growing technique (280-286). The substrate is first patterned with marker ink. Did they just draw a pattern on the substrate? And what kind of marker is used, water based/alcohol based? What is the color of the marker? Why does the ink suppress the nucleation of anthracene? A more detailed explanation on the procedure and mechanism will be helpful to reproduce the experiment.

We drew lines on glass slides by hands with an alcohol-based maker (KOKUYO, PM-41B). The marker color was blue, but we have not checked how the color affects the results. Much less anthracene crystals are nucleated on the marker ink than directly on glass, probably because the (100) plane of anthracene crystals has a large interfacial free energy with the ink, though the information on the dye is unavailable. We have added a comment on this marker ink in the supplementary information (page S3).

Summary of changes:

1. Page 2, the first sentence has been modified (line 27).
2. Page 3, the last sentence has been added (lines 85-87).
3. Page 10, the bottom panel of Figure 4d, xyz axes are defined.
4. Page 11, the caption of Figure 4f, description for top, middle, and bottom spectra is modified.
5. Page 14, the first paragraph right before “CONCLUSIONS” has been added (line 250-258).
6. Page 14, the last paragraph, the second last sentence has been added (line 276-277).
7. Page 16, the section for Data Availability has been added (line 319-322).
8. Page 17, the section for Completing Interests has been added (line 338-339)

Supplementary information

9. Page S3, the first paragraph, the third and fourth sentences are modified and added, respectively.
10. Page S4, the first paragraph, the last two sentences have been modified.
11. Page S7, the second paragraph, two sentences have been added at the end.
12. Page S8, Figure S6e and the corresponding caption have been added.
13. Page S9, the second last paragraph, the reference number is added for an exciton diffusion length of 700 nm.
14. Page S10, the second paragraph has been added.
15. Page S11, two figure panels (Figure S7e,f) have been added, and the corresponding figure caption has been added.
16. Page S14, the second paragraph has been added.
17. Page S15, three figure panels (Figure S10g-i) have been added, and the corresponding figure caption has been added.
18. Page S17, the second paragraph, the protocol to subtract the non-resonantly excited PL is made clearer by modifying the explanation and adding separated equations.
19. Page S18, in Figure S12d, an arrow has been added to indicate the position of interest.

REVIEWER COMMENTS

Reviewer #1 (Remarks to the Author):

The authors have significantly improved their manuscript (and the SI) and have addressed all my (little) concerns.

Therefore, I suggest to publish this work in Nature coms.

Reviewer #2 (Remarks to the Author):

The authors have addressed my comments in detail. I recommend the manuscript for publication

Response to reviewer report for manuscript NCOMMS-20-48650A by Otsuka *et al.*

Response Reviewer #1:

The authors have significantly improved their manuscript (and the SI) and have addressed all my (little) concerns.

Therefore, I suggest to publish this work in Nature coms.

We thank the reviewer for the valuable feedback that helped us to improve our manuscript. We are very happy to hear that the reviewer suggests the publication of our work in Nature Communications.

Response to Reviewer #2:

The authors have addressed my comments in detail. I recommend the manuscript for pulication

We thank the reviewer for taking his/her time to read our revised version of the manuscript. We are very happy to hear that the reviewer recommends our manuscript for publication in Nature Communications.